# Multi-Physical Field Simulation of Cracking during Crystal Growth by Bridgman Method

**DOI:** 10.3390/ma16083260

**Published:** 2023-04-20

**Authors:** Xinke He, Linnong Li, Xinqi He, Chao Xie

**Affiliations:** Faculty of Mechanical Engineering and Mechanics, Ningbo University, Ningbo 315211, China

**Keywords:** crystal growth, Bridgman method, numerical simulation

## Abstract

Crystal materials are prone to cracking during growth, which is a key problem leading to slow growth and difficulty in forming large-size crystals. In this study, based on the commercial finite element software COMSOL Multiphysics, the transient finite element simulation of the multi-physical field, including fluid heat transfer—phase transition—solid equilibrium—damage coupling behaviors, is performed. The phase-transition material properties and maximum tensile strain damage variables are customized. Using the re-meshing technique, the crystal growth and damage are captured. The results show the following: The convection channel at the bottom of the Bridgman furnace greatly influences the temperature field inside the furnace, and the temperature gradient field significantly influences the solidification and cracking behaviors during crystal growth. The crystal solidifies faster when it enters the higher-temperature gradient region and is prone to cracking. The temperature field inside the furnace needs to be properly adjusted so that the crystal temperature decreases relatively uniformly and slowly during the growth process to avoid crack formation. In addition, the crystal growth orientation also significantly affects the nucleation and growth direction of cracks. Crystals grown along the *a*-axis tend to form long cracks starting from the bottom and growing vertically, while crystals grown along the *c*-axis induce the laminar cracks from the bottom in a horizontal direction. The numerical simulation framework of the damage during crystal growth, which can accurately simulate the process of crystal growth and crack evolution and can be used to optimize the temperature field and crystal growth orientation in the Bridgman furnace cavity, is a reliable method to solve the crystal cracking problem.

## 1. Introduction

Organic optoelectronic functional materials, which have important application prospects in the fields of laser light sources, light-emitting diodes, scintillation counters, and microwave exciters [1,2], have garnered significant attention. The Bridgman method, proposed by the American physicist Bridgman in 1925 [3], has been widely used for the growth of organic materials, semiconductors, nonlinear optics, and many other functional crystals because of its versatility, low cost, and simplicity [4,5,6]. However, there is a fatal problem in the preparation of crystals. The cooling process of crystal growth is very likely to cause crystal cracking, which limits the development of such crystals toward large size, high integrity, and high quality.

Some modified Bridgman techniques have been proposed. For example, Arulchakkaravarthi et al. [7] grew anthracene single crystals using the Bridgman technique, and their study showed that optimizing the crystal growth furnace, by shortening the heating zone and increasing the growth zone for the furnace with two equal-length temperature zones, is conducive to avoiding crystal cracking and reducing lattice defects. Prabhakaran et al. [8] designed and manufactured an eight-temperature-zone crystal growth furnace, which dynamically changes the temperature gradient of the eight zones, enabling the vertical dynamic gradient solidification of the melt. They successfully grew 4-aminobenzophenone single crystals, and the experimental results showed that optimizing temperature gradient control and a vibration-free growth environment can improve the crystal quality and achieve better crystal performance.

However, the cracking behavior during crystal growth is closely related to mechanical stress, thermal stress, microstructure/defect evolution, and other factors [9,10,11,12,13,14,15], which is a complex multi-physical field coupling process that cannot be characterized in situ. Some scholars believed that the cracking behavior during the crystal growth process essentially originates from the difference in the thermal expansion coefficients within the crystal. Zhu et al. [9] investigated the cracking behavior during NaY(WO_4_)_2_ crystals growth and showed that the orientation of the seed crystals influences the cracking features of the crystals, and the thermal stress difference caused by the difference in the thermal expansion coefficients of the *z*-axis and *x*-axis of the crystals causes cracking directionality. When studying the production of CdSiP_2_ by the Bridgman method, Cheng [10] found that the *c*-axis of the crystal has a paradoxical thermal expansion coefficient, and he attributed the cracking to the anisotropy of thermal expansion within the crystal. Yang et al. [11] analyzed the thermal expansion anisotropy of CdSiP_2_ crystals using electronic structure theory. It was concluded that the excess valence electrons in the crystal strengthen the Si–P bond and suppress the thermal expansion of the crystal along the *c*-axis, causing the thermal expansion anisotropy. Liu et al. [12] discovered that thermal stress is the main cause of crystal cracking, and there are four types of crystal cracking: transverse lamellar fracture, longitudinal crack, cracking along the cleavage planes (101) or (011), and irregular fragmentation. Zhang et al. [13] found that residual tensile stress is the main cause of cracking in AlN single crystals grown on SiC seed crystals. Tanji et al. [14] investigated the effect of crystal growth direction on crystal cracking behavior and found that the difference in thermal expansion coefficients of PWO along the *a* and *c* directions is responsible for cracking. Shi et al. [15] found that the orientation of the as-grown crystal affects the crystal ingot cracking behavior. So far, the multi-physical mechanism of the cracking behavior during crystal growth is not clear enough.

Shyy et al. [16] introduced the computational methods and techniques for fluid flow with moving boundaries. Brown et al. [17,18,19] provided a detailed description of the growing method theory. Most existing studies are merely qualitative descriptions and quantitative analyses of growth stress, which cannot accurately predict crystal cracking. It is still far behind the accurate anti-cracking design. This study overlooks the flow and the fluid–solid coupling at the phase change interface and pays more attention to the damage caused by the residual stress and anisotropy in solid phase [20].

## 2. Materials and Methods

### 2.1. Theoretical Framework

The crystal growth process is a heat-transfer-controlled phase transition process. Because of the existence of temperature gradients, residual stress is generated during the solidification of the crystal and leads to damage. In this study, we perform transient finite element simulations for the multi-physical field coupling problem of heat transfer—phase transition—equilibrium—damage during the crystal growth process. The physical parameters of the crystal are shown in Table 1.

The purpose of this study is to develop a numerical simulation method that is not specific to a particular material. The method is built on the basis of multi-field coupling to physically capture the temperature-controlled crystal growth and thermal-stress-controlled cracking. Some parameters obtained from p-terphenyl are just for numerical calculation.

#### 2.1.1. Heat Transfer

During the crystal growth process, the heat transfer in the Bridgman furnace is mainly in the form of thermal convection and thermal conduction, and the transient heat transfer expression is given as follows [23]:(1)$$\rho C_{p}\frac{\partial T}{\partial t}+\rho C_{p}s\cdot \nabla T=\nabla \cdot (k\nabla T)+Q$$
where *ρ* is the density, *C*_p_ is the constant pressure heat capacity, *T* is the temperature, *t* is the time, *k* is the thermal conductivity, *Q* is the heat source, ***s*** is the velocity vector of liquid, and ∇ is the Hamilton operator.

#### 2.1.2. Phase Transition

The solid phase fraction *θ* is introduced in the simulation. Its continuous evolution is employed to describe the crystallization transition region [24]:(2)$$\theta =\left\{\begin{array}{cc} 0, & T>T_{m}+∆T\\ \frac{T-T_{m}+∆T}{2∆T}, & \left(T_{m}-∆T)\leq T\leq (T_{m}+∆T\right)\\ 1, & T<T_{m}-∆T \end{array}\right.$$
where *T*_m_ is the melting point and ∆*T* is the temperature half-width of the crystallization transition region.

The thermal conductivity of the crystal in the crystallization transition region is as follows:(3)$$k=\theta k_{l}+(1-\theta )k_{s}$$

The density of crystal in the crystallization transition region is as follows:(4)$$\rho =\theta \rho_{l}+(1-\theta )\rho_{s}$$

To convert the latent heat of phase transition into an equivalent specific heat, an indicator function for the solid–liquid phase fraction change can be defined as follows:(5)$$\alpha_{m}=\frac{1}{2}\frac{\theta \rho_{l}-(1-\theta )\rho_{s}}{\theta \rho_{l}+(1-\theta )\rho_{s}}$$

Assuming that the latent heat of phase transition is Δ*H*, the constant pressure-specific heat capacity of the crystallization transition region can be equated as follows:(6)$$C_{p}=\frac{1}{\rho }(\theta \rho_{l}C_{\mathrm{pl}}+(1-\theta )\rho_{s}C_{\mathrm{ps}})+\Delta H\frac{\partial \alpha_{m}}{\partial T}$$

#### 2.1.3. Solid Equilibrium

The crystals in the Bridgman furnace are linearly elastic materials after solidification. The constitutive equation is as follows:(7)$$\sigma \;=C:\varepsilon^{e}$$
where σ is the stress tensor, C is the stiffness tensor, $\varepsilon^{e}$ is the elastic strain tensor.

The solid movement equation is given by the following:(8)$$\rho \frac{\partial^{2}u}{\partial t^{2}}=\nabla \cdot \sigma$$
where *u* is the displacement.

The thermal strain tensor $\varepsilon^{t}$ can be defined as follows:(9)$$\varepsilon^{t}=\mu d_{T}$$
where *μ* is the anisotropic thermal expansion coefficient tensor and $d_{T}$ is the difference in temperature.

The small deformation strain tensor ε can be defined as follows:(10)$$\varepsilon =\frac{1}{2}(\nabla u+u\nabla ),$$
and it can be decomposed into thermal and elastic components:
(11)$$\varepsilon =\varepsilon^{t}+\varepsilon^{e}$$

The crystals are set up as a body-centered tetragonal structure with significant anisotropy, and the coefficient of thermal expansion in the *a*-direction is only half of that in the *c*-direction. The anisotropic thermal expansion coefficient matrix is introduced to set the thermal expansion anisotropy, and the anisotropy of stiffness is ignored. The growth behavior of crystals along different orientations is studied by defining different anisotropic thermal expansion coefficient matrices:

When the crystal grows along the *a*-axis, the anisotropic thermal expansion coefficient matrix is set to $\mu_{1}=\left(\begin{array}{ccc} \beta & 0 & 0\\ 0 & \frac{\beta }{2} & 0\\ 0 & 0 & \frac{\beta }{2} \end{array}\right)$; when the crystal grows along the *c*-axis, $\mu_{2}=\left(\begin{array}{ccc} \frac{\beta }{2} & 0 & 0\\ 0 & \beta & 0\\ 0 & 0 & \frac{\beta }{2} \end{array}\right)$.

#### 2.1.4. Damage

Based on the criterion of maximum tensile strain, the maximum value of elastic tensile strain is used to determine whether a material is cracked and where the fracture occurs.

Define the step damage variable *M*,
when $\varepsilon_{11}^{e}\geq \varepsilon_{22}^{e}$:
(12)$$M=\left\{\begin{array}{c} 1,\;\;\varepsilon_{11}^{e}<1.2\times 10^{-4}\\ 0,\;\;\varepsilon_{11}^{e}\geq 1.2\times 10^{-4} \end{array}\right.,$$when $\varepsilon_{11}^{e}<\varepsilon_{22}^{e}$:(13)$$M=\left\{\begin{array}{c} 1,\;\;\varepsilon_{22}^{e}<1.2\times 10^{-4}\\ 0,\;\;\varepsilon_{22}^{e}\geq 1.2\times 10^{-4} \end{array}\right.,$$
where $\varepsilon_{11}^{e}$ and $\varepsilon_{22}^{e}$ are the two normal components of the elastic strain tensor in the X and Y directions, respectively. 

The degradation of Young’s modulus should be considered as follows:(14)$$E=E_{\mathrm{crystal}}\theta M$$

The max function is a built-in function in COMSOL Multiphysics to determine the maximum between two components. In this study, it is defined as follows:(15)$$m_{t}=\max(\varepsilon_{11}^{e},\varepsilon_{22}^{e})$$
where $m_{t}$ represents the maximum between the normal elastic strains $\varepsilon_{11}^{e}$ and $\varepsilon_{22}^{e}$.

The threshold value of the elastic strain is hard to be found. A reasonable value can be obtained according to the fracture toughness calculation:(16)$$K_{\mathrm{IC}}=E_{\mathrm{crystal}}\varepsilon_{c}\sqrt{\pi a}$$
where $\varepsilon_{c}=\;1.2\times 10^{-4}$ is the critical strain, and *a* is the critical crack length.

According to Formula (16), the calculated fracture toughness, which belongs to the brittle material grade, is 2.3 MPa∙m^1/2^. The fracture toughness is much lower than that of steel and is close to those of other grown crystals [25].

At the material point where the maximum elastic strain is less than the threshold value of 1.2 × 10^−4^, *M* = 1, and Young’s modulus remains unchanged. At the material point where the maximum elastic strain is greater than or equal to 1.2 × 10^−4^, *M* = 0, Young’s modulus also degrades into 0, and the crack is formed.

### 2.2. Numerical Simulation

In this study, the damage during crystal growth is modeled and simulated based on the commercial finite element software COMSOL Multiphysics.

Based on the structure of the Bridgman growth system, the geometric model shown in Figure 1 is established. The longitudinal section of the Bridgman growth system is set as the modeling object; meanwhile, the secondary factors such as the lifting device and the insulation cover are ignored. The two-dimensional models of the furnace cavity and the quartz bottle are established.

The furnace cavity, with an inner diameter of 80 mm, is 700 mm long, and the quartz bottle is 20 mm in diameter and 200 mm in length. The furnace heating system is equivalent to the upper and lower heating wires to supply heat separately, forming an axially unidirectional decreasing temperature gradient zone. The lift channel at the bottom of the furnace cavity is retained for heat exchange with the environment during crystal growth. To compare the effects of different temperature gradients on crystal growth, three sets of models are established. The positions of the heating wires in Figure 1a,b are the same, and only the lower heating wire temperatures are different. While the heating wires in Figure 1c are kept at the same temperatures as in Figure 1b, the positions of the heating wires and quartz bottle are lifted to reduce the influence of the lift channel on the temperature gradient. The three different processing technologies are defined as process A, B, and C.

The imposed boundary conditions are as follows: the furnace cavity boundary is set to be thermally insulated; the lift channel is set for the convective heat flux which exchanges with the environment; the convection coefficient *h* = 10 W/(m^2^∙K); the quartz bottle wall is set to be a conductive thin layer.

The initial conditions are as follows: the velocity of the quartz bottle is 0 mm; the ambient temperature is 293.15 K, the temperature inside the furnace is 505 K, and the pre-melting state of the crystal is simulated.

Considering the regular structure of the established two-dimensional model, COMSOL Multiphysics is employed for meshing. The linear elements are set, the maximum element size is set to 3.34 mm, and the minimum is 0.2 mm. The meshing of the three processes is similar, and only the meshing schematic of process A is shown in Figure 2. The automatic re-meshing is set in the solver, the condition for re-meshing is the meshing quality, and the threshold is set to 0.2.

The analysis is solved in two steps. Transient fully coupling direct solution for 5000 s is used in step 1 to simulate the furnace cavity from pre-melting temperature to working temperature, and the quartz bottle is kept stationary during this analysis step so that a steady state of temperature is reached and maintained in the furnace cavity. Step 2 uses a transient iterative separation method to solve the temperature, displacement field, and spatial nodal displacements. The temperatures of the heating wires keep constant, and the quartz bottle moves vertically downward at a uniform speed of 1 mm/h. The movement of the quartz bottle in the temperature gradient field, the process of solidification, deformation, and damage of the crystal are simulated.

## 3. Results

Figure 3 shows the simulation contour plot at the initial moment of the crystal phase transition for processes A, B and C. The contour plot inside the quartz bottle indicates the solid phase fraction of the crystal (white for the complete solid phase, light blue for the complete liquid phase), the contour plot outside the quartz bottle indicates the temperature distribution (red for high temperature, blue for low temperature), the isotherms represent the temperature gradient inside the furnace cavity, and the rectangular wire frames indicated by the black arrows in the figure are the initial locations of the quartz bottles. It can be observed from the figure that the phase transition occurs first at the bottom of the bottle (the positions indicated by the dark blue arrows), where the solid phase fraction *θ* is equal to 1, and the crystals change from the liquid phase to the solid phase. The moments of phase transition initiation of the crystals for processes A, B, and C are 70,000 s, 125,000 s, and 140,000 s, respectively. The three different Bridgman furnace crystal growth processes can cause different temperature field distributions. The solidification of crystal is very sensitive to the temperature gradient, therefore when the crystal enters the region of the high temperature gradient, the rapid drop of temperature can significantly promote the solidification process of the crystal (process A). When the temperature gradient in the furnace cavity is reduced, the solidification process of the crystals becomes slower (processes B and C).

### 3.1. Crystal Growth along the a-Axis

Figure 4 shows the crystal growth of the three processes at different moments when the crystals grow along the *a*-axis. The internal contour plot of the quartz bottle shows the Young’s modulus of the crystal (black represents a Young’s modulus of 55 GPa, and white represents a Young’s modulus of 0 GPa), the external contour plot of the quartz bottle shows the temperature distribution (red represents high temperature, and blue represents low temperature), and the isotherms represent the temperature gradient inside the furnace cavity. The white part of the upper part of the quartz bottle with a Young’s modulus of 0 GPa means that the crystal is still in the liquid state; the black area at the lower end represents a crystal with a Young’s modulus of 55 GPa. The crystal has undergone a phase transition and transformed into the solid state. It can be observed from Figure 4a,d,e,g,h that white vertical bands appear at the bottom of the crystal (the positions indicated by the blue arrows), and the Young’s modulus of the material point drops to 0 GPa, which means that vertical cracks nucleate and grow at the bottom of the crystal.

Figure 4a–c show the crystal growth in the three processes at 160,000 s. For process A, the crack nucleates at the bottom of the crystal, where the solidified part is about 87 mm in length, and the length of the crack nucleation is about 40 mm. The crack accounts for 47.6% of the solidified crystal length. For process B, the solidified part of the crystal is about 34 mm in length, and no crack appears. For process C, the solidified part is 17 mm in length, and no crack appears. Figure 4d–f show the growth of the crystal in the three processes at 260,000 s. The solidified part of the crystal for process A reaches about 153 mm, and the crack is about 107 mm, accounting for 69.9% of the solidified crystal length. A crack nucleates in the crystal for process B, and the length of the solidified part of the crystal is about 95 mm. The initial nucleation of the crack is 38 mm, and the crack accounts for 40% of the solidified crystal length. For process C, the length of the crystal reaches about 80 mm, and still no crack appears. Figure 4g–i show the crystal growth in the three processes at 275,000 s. The solidified part of the crystal for process A is about 160 mm in length, and the length of the crack is about 113 mm, accounting for 70.6% of the length of the solidified crystal. For process B, the length of solidification is about 110 mm, and the length of the crack is about 59 mm, and the crack accounts for 53.6% of the crystal length. For process C, the length of the solidified part of the crystal also reaches about 95 mm, but still no crack appears.

### 3.2. Crystal Growth along the c-Axis

Figure 5 shows the crystal growth for the three processes at different moments when the crystals grow along the *c*-axis. Same as in Figure 4, the internal contour plot of the quartz bottle shows the Young’s modulus of the crystal (black represents a Young’s modulus of 55 GPa, and white represents a Young’s modulus of 0 GPa), the external contour plot of the quartz bottle shows the temperature distribution (red represents high temperature, and blue represents low temperature), and the isotherms represent the temperature gradient inside the furnace cavity. It can be observed from Figure 5a,d,e,g,h that white horizontal bands appear at the bottom of the crystals (the positions indicated by the blue arrows), and the Young’s modulus of the material point drops to 0 GPa, which means that horizontally oriented lamellar cracks nucleate and grow at the bottom of the crystal. 

Figure 5a–c show the crystal growth in the three processes at 130,000 s. For process A, the crack nucleates at the bottom of the crystal, where the solidified part is about 65 mm in length, and the thickness of the damage layer is about 3 mm. The damage layer accounts for 4.6% of the solidified crystal length. For process B, the solidified part of the crystal is about 7 mm in length and no crack appears. For process C, no solidification occurs. Figure 5d–f show the crystal growth in the three processes at 210,000 s. For process A, the length of the solidified part is about 131 mm, the thickness of the damage layer reaches about 77 mm, accounting for 58.7% of the solidified crystal length. A crack nucleates in the crystal for process B, and the length of the solidified part of the crystal is about 66 mm, the thickness of the initial nucleation of the crack is about 3 mm, and the damage layer accounts for 4.5% of the solidified crystal length. For process C, the length of the solidified crystal reaches 65 mm, and still no crack appears. Figure 5g–i show the crystal growth in the three processes at 215,000 s. The solidified part of the crystal for process A is about 133 mm in length, and the thickness of the damaged layer is 79 mm, accounting for 59.4% of the length of the solidified crystal. For process B, the length of solidification is about 68 mm, the thickness of the damage layer is still about 3 mm, and the thickness of the damage layer accounts for 4.4% of the crystal length. The length of the solidified part of the crystal for process C also reaches about 66 mm, but still no crack appears.

## 4. Discussion

### 4.1. Maximum Tensile Strain Criterion and Crack Shape

The residual stress generated in the crystal during the cooling and solidification process is a no-shear stress state. The crystal is anisotropic; its damage exhibits anisotropic brittle damage behavior, and the maximum tensile strain criterion is applicable. The material point is no longer capable of bearing loads after the fracture occurs. The Young’s modulus of the damaged material point in the model should be reduced to 0, leading to the stress concentration at the crack tip.

Figure 6 shows the crystal growth along *a*-axis in the process A at 260,000 s. The contour lines represent the Young’s modulus of the crystal (the red line represents the zero value for the completely damaged crystal, and the light blue represents the original value for the intact crystal). It can be seen that the completely damaged area with zero modulus, namely the crack, is very narrow. The rectangle area can be defined as the damaged area instead of a crack.

### 4.2. Effect of Temperature Gradient on Crystal Growth Process

Figure 7 shows the distribution of the temperature field in the Bridgman furnace for processes A, B, and C. The rectangular wire frames indicated by the blue arrows in the figure are the locations of the quartz bottles. It is shown that the isotherms in the low temperature zone close to the bottle of the furnace cavity for process A are the densest, which indicates that the temperature gradient in this process is the largest. For process B, the temperature of the lower heating wire is raised, so that the temperature gradient is reduced. However, the bottom of the quartz bottle is close to the lift channel and obviously influenced by thermal convection, causing the temperature gradient around the bottom of the quartz bottle still to remain great. For process C, the quartz bottle is farther away from the lift channel, and the temperature gradient in the crystallization region is slightly affected by thermal convection.

Comparing the crystal growth under the three processes, it can be concluded that for process A, the crystal cools down the fastest, solidifies the earliest, and is prone to fracture. For process B, by changing the temperatures of the heating wires, the temperature gradient is relatively lesser, which slows down the cooling rate of the crystal, reduces the thermal strain and residual stress, and delays the fracture behavior. For process C, in addition to changing the temperatures of the heating wires, the positions of the heating wires and the quartz bottle are adjusted to decrease the temperature gradient. Compared to process B, process C loses a small amount of efficiency but greatly improves the crystal integrity.

In the engineering application of the Bridgman method for crystal growth, the temperature field in the furnace cavity needs to be optimized so that the temperature gradient in the crystallization region is less. Crack is less likely to nucleate when the crystal is located in the region where the temperature drops slowly and uniformly. The temperature field in the furnace cavity can be controlled by varying the temperatures of the heating wires. In addition, to reduce the influence of heat convection on the crystal growth, it is necessary to keep the crystal far away from the lift channel at the bottom of the furnace cavity or to improve the air-tightness of the lift channel. Selecting a suitable process can effectively reduce the cracking phenomenon during the Bridgman crystal growth process and greatly improve the crystal integrity.

### 4.3. Anisotropic Crystal Growth Process along Different Orientations

When the crystal grows along the *a*-axis, a crack initiates from the bottom when $\varepsilon_{11}^{e}$ reaches a threshold value of 1.2 × 10^−4^. According to the maximum tensile strain criterion, the crack then propagates along the vertical direction as a result of the greatest normal strain component $\varepsilon_{11}^{e}$ at the crack tip. When the crystal grows along the *c*-axis, it produces horizontal lamellar cracks initiating from the bottom owing to the greatest normal strain component $\varepsilon_{22}^{e}$ within the crystal. Comparing two processes, when the crystal grows along the *a*-axis, the crack appears later but grows more rapidly and the length of the crack accounts for a larger proportion of the solidified crystal length; when the crystal grows along the *c*-axis, the crack appears earlier; however, the damage layer accounts for a smaller proportion of the solidified crystal. The crystals with better integrity can be prepared by reasonably adjusting the crystal growth orientation.

The simulation results of crystal fracture behavior are in good accordance with the classical experimental results [12,13,14,15].

## 5. Conclusions

In this study, a multi-physical field numerical simulation framework is built for the damage during crystal growth processes. Based on the commercial finite element software COMSOL Multiphysics, fluid heat transfer—solidification phase transition—solid equilibrium—damage are sequentially coupled. The maximum-tensile-strain brittle damage criterion, together with the stiffness recession function, is self-defined. Using the re-meshing technique, the solidification, deformation, and damage processes of crystal are accurately simulated. The following conclusions were obtained:(1)The influence of the temperature gradient field on the solidification and cracking behaviors during crystal growth is of great significance. In this study, three different temperature fields were constructed for three different processes. When the crystal grows along the *a*-axis, the solidified parts of the crystal for processes A, B, and C reach 160 mm, 110 mm, and 95 mm at 275,000 s, respectively. In addition, the crack accounts for 70.6% for A, 53.6% for B, and 0% for C. When the crystal grows along the *c*-axis, the solidified parts for processes A, B, and C reach 133 mm, 68 mm, and 66 mm at 215,000 s, respectively. The crack accounts for 59.4% for A, 4.4% for B, and 0% for C. By adjusting the temperature gradient, the crystal temperature decreases relatively uniformly and slowly during the growth process, thus avoiding the formation of cracks. At the same time, the lift channel at the bottom of the Bridgman furnace greatly influences the temperature field inside the cavity due to the heat convection with the environment. When the furnace is working, the crystal growth area should be kept away from the lift channel at the bottom of the furnace cavity, or the air-tightness of the furnace cavity should be improved to reduce the influence of heat convection on the temperature field inside the furnace cavity. In engineering applications, the temperature of the Bridgman furnace cavity needs to be properly optimized so that the temperature of the crystal drops steadily and uniformly during the growth process. This can slightly reduce the efficiency of the crystal growth, but can greatly improve the crystal integrity.(2)The crystal produces thermal shrinkage during solidification, and the bottom cools down the fastest, causing the maximum residual stress and elastic strain. Due to the anisotropy of the thermal expansion, when the crystals grow along different orientations, the maximum elastic strain is generated in different directions, leading to crack nucleation and growth along different directions. Crystals grown along the *a*-axis tend to form long cracks starting from the bottom and growing vertically, while crystals grown along the *c*-axis induce the laminar cracks from the bottom in a horizontal direction. Cracks appear earlier when the crystals grow along the *c*-axis, whereas the damage layer accounts for a smaller proportion of the solidified crystals. Crystal integrity can be effectively safeguarded by reasonable adjustment of the orientation of crystal growth.

The crystal damage numerical simulation framework established in this study restores the coupling process of multiple physical fields during crystal growth except for the flow and liquid–solid interface interaction, which can replace the in situ characterization and monitoring techniques to a certain extent, and quantitatively optimize the process of the Bridgman method for preparing the crystal with anti-cracking ability.

## Figures and Tables

**Figure 1 materials-16-03260-f001:**
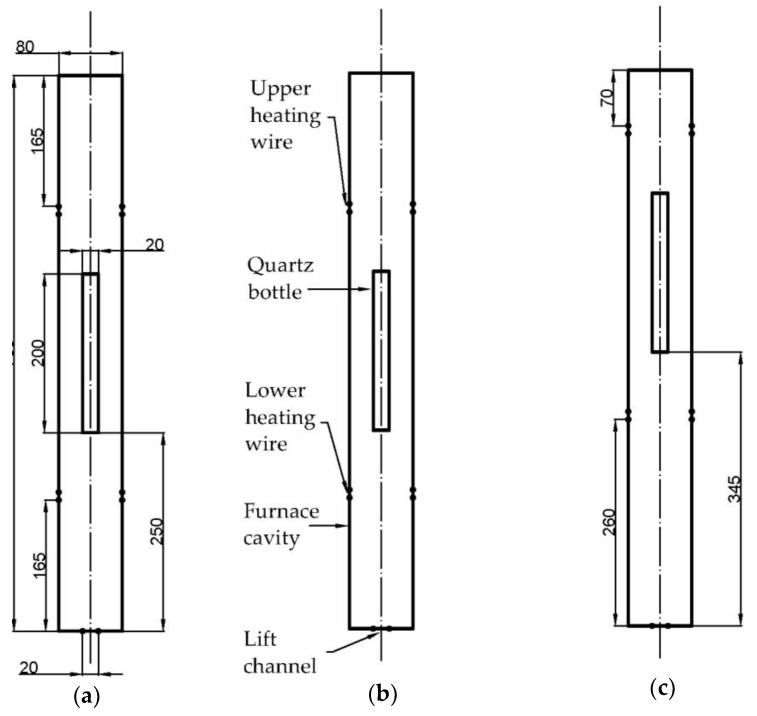
Geometric model of Bridgman growth system: (**a**) process A: upper heating wire 501 K, lower heating wire 460 K; (**b**) process B: upper heating wire 501 K, lower heating wire 480 K; (**c**) process C: upper heating wire 501 K, lower heating wire 480 K.

**Figure 2 materials-16-03260-f002:**
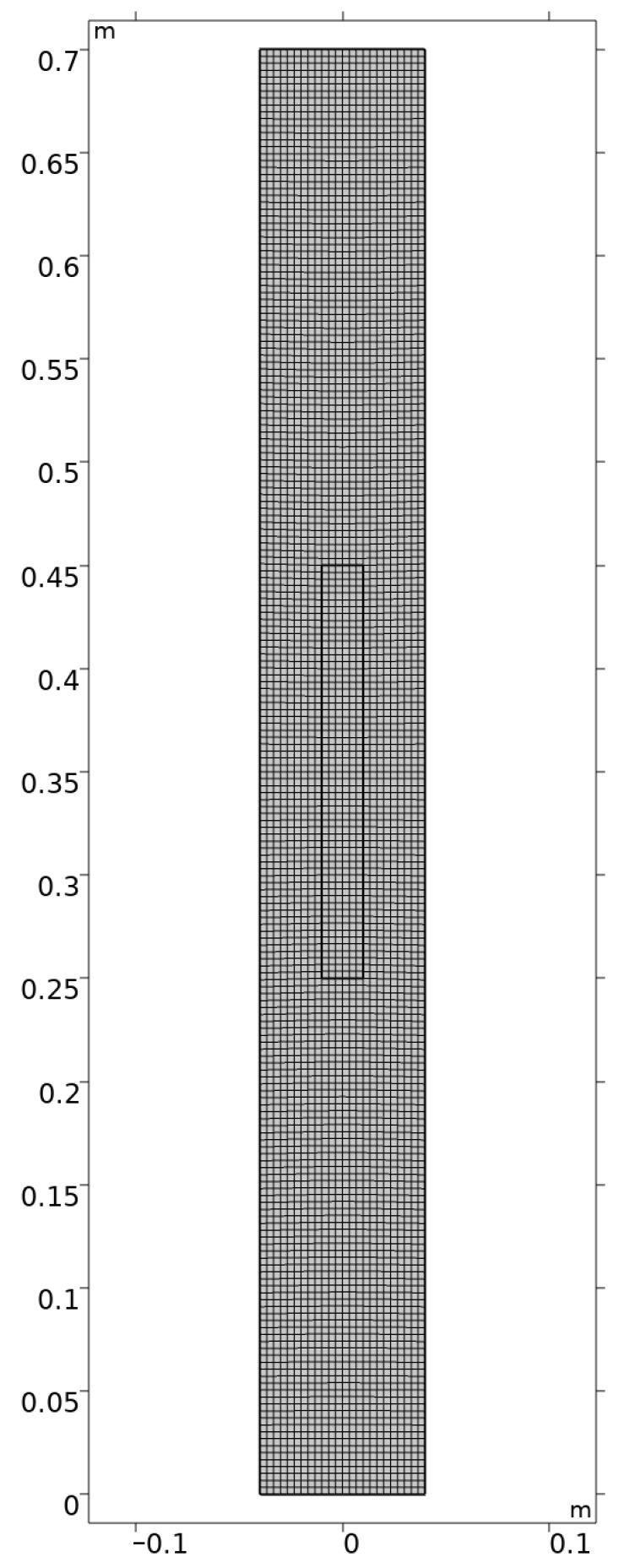
The meshing of Bridgeman Furnace for Process A.

**Figure 3 materials-16-03260-f003:**
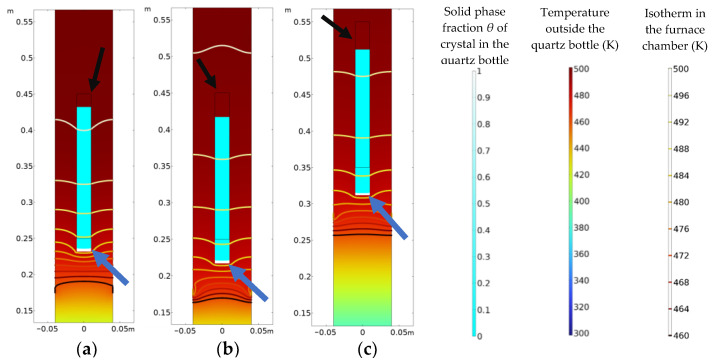
Initial moments of the phase transition: (**a**) process A; (**b**) process B; (**c**) process C.

**Figure 4 materials-16-03260-f004:**
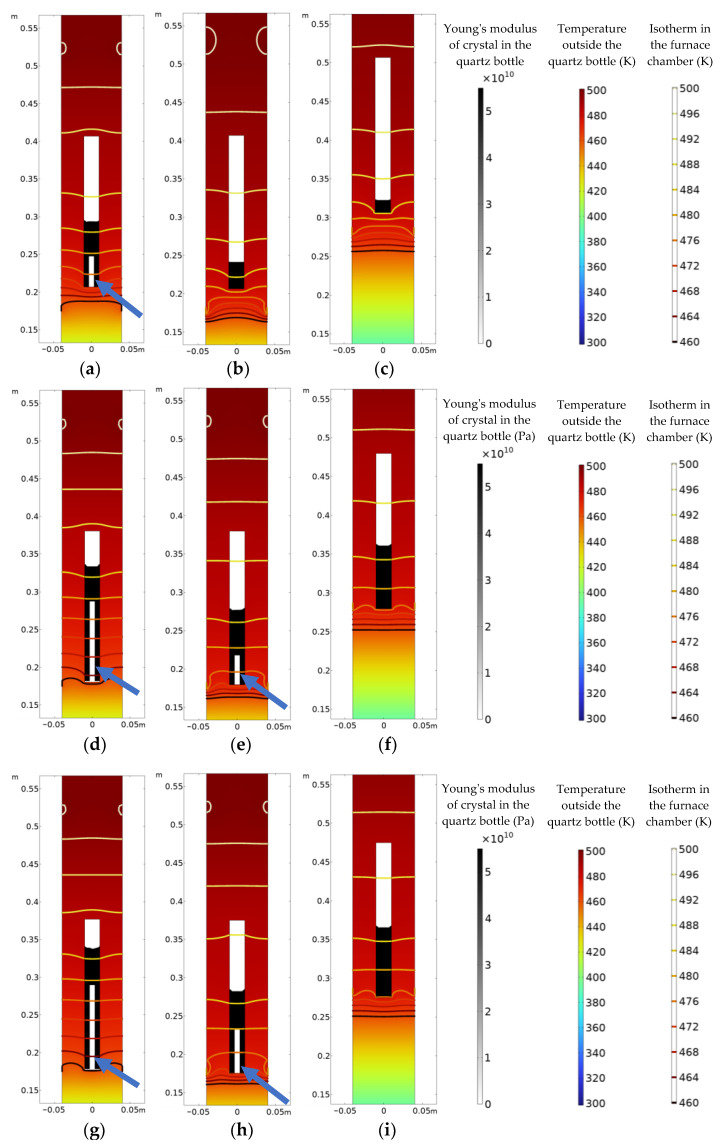
Crystal growth along the *a*-axis: (**a**–**c**) 160,000 s; (**d**–**f**) 260,000 s; (**g**–**i**) 275,000 s; (**a**,**d**,**g**) for process A; (**b**,**e**,**f**) for process B; (**c**,**f**,**i**) for process C.

**Figure 5 materials-16-03260-f005:**
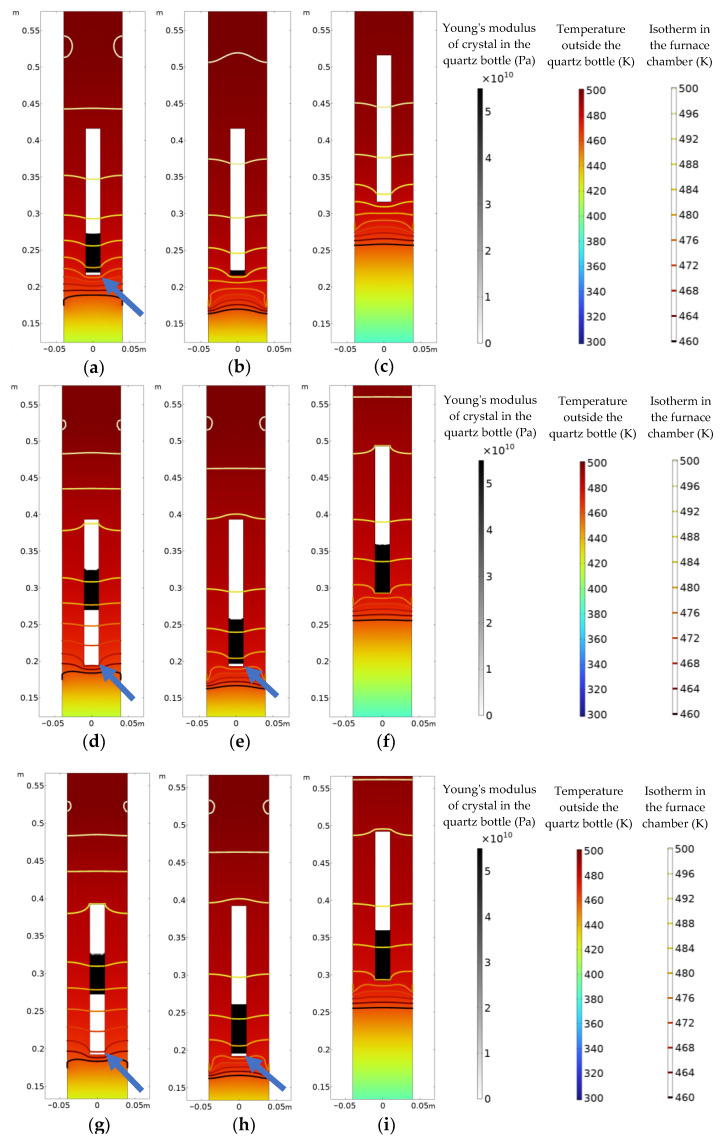
Crystal growth along the *c*-axis: (**a**–**c**) 130,000 s; (**d**–**f**) 210,000 s; (**g**–**i**) 215,000 s; (**a**,**d**,**g**) for process A; (**b**,**e**,**f**) for process B; (**c**,**f**,**i**) for process C.

**Figure 6 materials-16-03260-f006:**
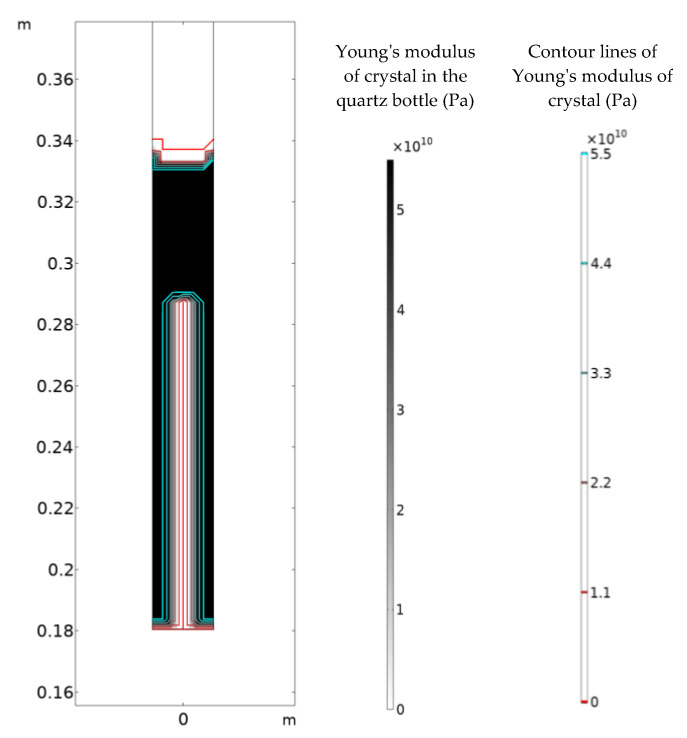
The internal contour plot of the quartz bottle.

**Figure 7 materials-16-03260-f007:**
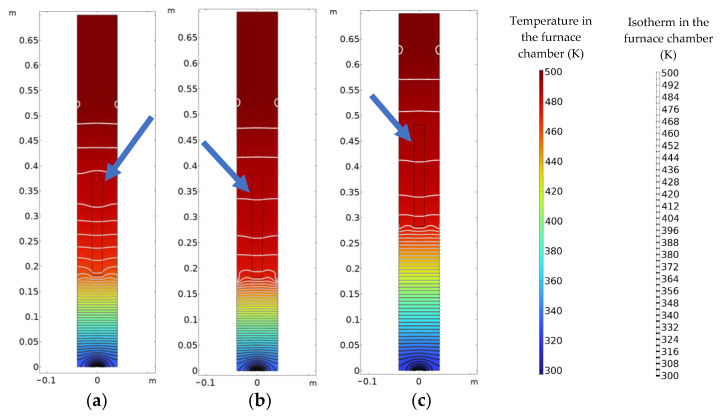
Temperature field distribution at 250,000 s in Bridgman furnace: (**a**) process A; (**b**) process B; (**c**) process C.

**Table 1 materials-16-03260-t001:** Physical parameters [21,22].

Physical Parameters	Numerical Value	Unit
Young’s modulus	*E*_crystal_ = 55,000	MPa
Poisson’s ratio	*v* = 0.33	-
Density (solid state)	*ρ*_s_ = 962.4	kg/m^3^
Density (liquid state)	*ρ*_l_ = 962.4	kg/m^3^
Thermal conductivity (solid state)	*k*_s_ = 0.12866	W/(m∙K)
Thermal conductivity (liquid state)	*k*_l_ = 0.12866	W/(m∙K)
Specific heat capacity (solid state)	*C*_ps_ = 2001.704	J/(K∙kg)
Specific heat capacity (liquid state)	*C*_pl_ = 2112.373	J/(K∙kg)
Melting point	*T*_m_ = 486	K
Latent heat of melting	Δ*H* = 153.27	kJ/kg
Coefficient of thermal expansion	*β* = 8.8 × 10^−4^	1/K

## Data Availability

The main data supporting the findings of this study are available within the article. Extra data are available from the corresponding author upon reasonable request.

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
