# Peer review of "Multi-Physical Field Simulation of Cracking during Crystal Growth by Bridgman Method"

_materials, 2023, doi:10.3390/ma16083260_

Round 1
Reviewer 1 Report
The manuscript concerns modelling with a specific crystal used as an example, so the introduction should have focused on the p-terphenyl crystal and the problems of its growth by the Bridgman method. It is unclear why so many references refer to nonlinear optical crystals, including KDP (I counted sixteen), which are grown by a completely different method, namely from an aqueous solution. The p-terphenyl crystal is considered a substance relevant to functional electronics and not to nonlinear optics. There is also a lack of citations on the growing method and the main properties of the crystal.
In general, the references cited in the manuscript do not properly cover the recent scientific research in this direction. The majority of works are by authors from PR China, except 1 (407), 20 (443, with improper writing of names) and 21 (445). It was essential to turn to fundamental works here, which describe the theory of the growing method in detail. In particular, the book Shyy, Wei, H. S. Udaykumar, and Madhukar M. Rao. Computational fluid dynamics with moving boundaries. CRC Press, 1995. The articles by Brown and Chang should also be recalled:
https://doi.org/10.1016/0022-0248(74)90161-4
https://doi.org/10.1002/aic.690340602
https://doi.org/10.1016/0022-0248(83)90225-7 )
Overall, such kind of research is more suitable for the journal “Crystals”, where an article on the growth of pentacene-doped p-terphenyl crystal by the Bridgman method was recently published (https://doi.org/10.3390/cryst13010002).
The theoretical part is described too briefly, there are major errors, e.g. "∇ is the Hamiltonian operator" (121), while it probably refers to the nabla symbol, which denotes three distinct differential operators: the gradient, the divergence, and the curl.
It is unclear how the terms "crystallographic transition region" (128) and "crystallization transition region" (130) differ. Figure 2 is not referenced anywhere in the text. At the same time, there are references to figures 4, 5, 6, but they are absent in the captions (with duplicated numbers instead - figures 1, 2, 3).
It makes little sense to analyze the manuscript more thoroughly since its formatting is quite disorganized, so it cannot be accepted in its current form.
Reviewer 2 Report
Comments
I have read the manuscript, “Multi-physics field numerical simulation of cracking behavior
during nonlinear optical crystal growth” submitted to the “Materials” Journal. The authors investigated the modeling and simulating of the crystal growth for nonlinear optical crystal materials in order to eliminate/understand crack evolution. They used COMSOL Multiphysics software. The manuscript is interesting and includes new approaches in the field of crystal growth. I think it can be accepted after minor revisions.
-Please improve the references and uses more actual references.
-Please add a reference for the first paragraph of the Introduction.
- I recommend detailing why these parameters are selected and most importantly their values/ranges.
-I strongly advise to improve English of manuscript.
-If possible, I suggest adding real experimental data/results (at least one) using simulation results to show consistency/reliability in the simulation.
- Add quantitative results for the conclusion part.
Reviewer 3 Report
The paper submitted for peer review, entitled «Multi-physics field numerical simulation of cracking behavior during nonlinear optical crystal growth», is an attempt to obtain a digital simulation of the crystal growth processes of organic nonlinear materials such as p-terphenyl and tetracene. The data presented by the authors may be useful to some researchers who practice the Bridgman method for growing single crystals. At the same time, it should be noted that the Bridgman method (as well as its modifications - the Tamman method, the Obreimov-Shubnikov method, and the horizontal crystallization by the Bagdasarov method) is used very widely for a very large number of materials both in experimental research and in industrial crystal growth. It is not very clear from the information and conclusions presented in the article can be useful specifically for practical applications. Also, some of the parameters given by the authors for modeling are very difficult or impossible to control in the real process of the crystal growth.
The main disadvantages of this article are listed below:
1. The title of the article is too general. A more specific name is required, indicating the method and material to be crystallized.
2. The annotation should also indicate the type of crystals that were modeled. In the abstract, the authors talk about axes a and c, where cracking occurs, but this is not correct for all crystalline materials.
3. In the introduction, a significant part of the text is devoted to the description of the processes of growth and damage of KDP crystals, and references to these crystals make up about half of the total (refs. 13-22). Why do the authors pay so much attention to KDP crystals? (grown by a completely different method from low-temperature aqueous solutions) and having nothing in common with the Bridgman method. At the same time, the mechanism and reasons for the formation of cracks in KDP crystals apparently also differ greatly.
4. It is required to clarify why, as an example, modeling is given specifically for p-terphenyl crystals, which have an very low melting point compared to other compounds grown by this method.
5. The basis of the Bridgman method is the creation of nucleation in the melt in the lower narrow “neck” of the ampoule and the subsequent geometric selection of seeds during the movement of the temperature gradient. The authors do not take this fact into account .
6. In the conclusions, the recommendation to reduce the temperature gradient is very strange for the Bridgman method, since it is this method that implies the presence of a sharp and significant temperature drop in the working zone of the furnace, for which multi-zone furnaces and special screens are used.
In my opinion, the article needs major revision in order to correct the material presented and show the main idea of the authors of this article.
Round 2
Reviewer 3 Report
Thanks for the replies to my comments. After this correction, in my opinion, the article can be accepted for publication in the "Materials".
Reviewer 4 Report
The authors improved their manuscript and answered my comments properly.
I recommend the Accept in present form.